# Contributions of Women in Recent Research on Biopolymer Science

**DOI:** 10.3390/polym14071420

**Published:** 2022-03-30

**Authors:** Unnimaya Thalakkale Veettil, Sheila Olza, Nelly Brugerolle de Fraissinette, Elodie Bascans, Natalia Castejón, Amandine Adrien, Rut Fernández-Marín, Corinne Nardin, Susana C. M. Fernandes

**Affiliations:** 1E2S UPPA, CNRS, IPREM, Universite de Pau et des Pays de l’Adour, 64000 Pau, France; unnimayatv0@gmail.com (U.T.V.); sheila.olza@univ-pau.fr (S.O.); nelly.brugerolle-de-fraissinette@univ-pau.fr (N.B.d.F.); elodie.bascans@univ-pau.fr (E.B.); natalia.castejon@univie.ac.at (N.C.); amandine.adrien@univ-pau.fr (A.A.); corinne.nardin@univ-pau.fr (C.N.); 2E2S UPPA, Marine Materials Research Group, Universite de Pau et des Pays de l’Adour, 64600 Anglet, France; 3Department of Cellular Biology and Histology, Faculty of Medicine Nursery, University of the Basque Country, B Sarriena, s/n, 48940 Leioa, Spain; 4CNRS, INRAE, INSA, Toulouse Biotechnology Institute (TBI), Université de Toulouse, 31400 Toulouse, France; 5Environmental and Chemical Engineering Department, University of the Basque Country UPV/EHU, Plaza Europa 1, 20018 Donostia-San Sebastián, Spain; rfermarin@gmail.com

**Keywords:** natural polymers, agar, chitin, chitosan, cellulose, collagen, marine polysaccharides, marine CAZymes

## Abstract

Nowadays, biopolymers are playing a fundamental role in our society because of the environmental issues and concerns associated with synthetic polymers. The aim of this Special Issue entitled ‘Women in Polymer Science and Technology: Biopolymers’ is highlighting the work designed and developed by women on biopolymer science and technology. In this context, this short review aims to provide an introduction to this Special Issue by highlighting some recent contributions of women around the world on the particular topic of biopolymer science and technology during the last 20 years. In the first place, it highlights a selection of important works performed on a number of well-studied natural polymers, namely, agar, chitin, chitosan, cellulose, and collagen. Secondly, it gives an insight into the discovery of new polysaccharides and enzymes that have a role in their synthesis and in their degradation. These contributions will be paving the way for the next generation of female and male scientists on this topic.

## 1. Introduction

Natural polymers, also known as biopolymers, are naturally occurring materials, formed during the life cycles of living organisms. They can be derived from plants (e.g., cellulose), algae (e.g., agar), animals (e.g., chitin and collagen), bacteria (e.g., bacterial cellulose), and fungi (e.g., chitosan) (Figure 1). Biopolymers present unique properties and advantages, namely, high biocompatibility, lack of toxicity, biodegradability, stability, available functional groups, and often are low cost. These properties make natural polymers interesting sustainable alternatives to replace synthetic polymers in materials development. Thus, in the last decades, there has been a prompt development and breakthrough in biopolymer science and technology to better understand their fundamental and applied biological, physicochemical, morphological, and mechanical properties [1,2,3,4,5,6,7]. Moreover, the challenging prospection for novel natural polymers has led to fundamental discoveries; in particular, within the rich fount of marine polysaccharides, and their related enzymes (carbohydrate-active enzymes or CAZymes) that represent marine microscopic life [8,9,10].

Henri Braconnot or Anselme Payen are names that are very well known in the biopolymer field for their outstanding discovery of chitin and cellulose. Maybe the names Angelina Fanny Hesse and Wanda Kirkbride Farr are not as widely recognized as others are in the field, however, the outstanding idea of Hesse to use agar as culture media for growing microorganisms in 1882 revolutionized microbiology [11]. In 1940, Farr discovered the mechanism of cellulose formation in the plant cell walls, answering a question that had puzzled scientists for a long time [12]. Since the discovery of these polymers, a number of great works have highlighted their importance and greatly implemented the work of the pioneers of biopolymer science. During the last 20 years, many research teams have dedicated their time to biopolymer sciences and have led to several advances, discoveries, and innovations (Figure 2), building upon the general knowledge of natural polymers [13]. In this paper, we wanted to showcase some of these recent devoted efforts by highlighting the work of women investigators worldwide (both well-recognized leaders and rising stars) in the biopolymer community. The aim of this paper is not to list all the numerous accomplishments of women in the biopolymer science as it would be an impossible task. Herein, we rather want to highlight some examples of scientific advances of some well-known natural polymers that have been made possible by the dedication of researchers in women-led teams.

## 2. Advances in Biopolymer Research: Examples from Well-Known Biopolymers

### 2.1. Agar

The use of agar as a gelling agent is very ancient, and in particular, in several Southeast Asian cuisines. It is said that it was discovered in the 17th century in Japan by an innkeeper named Mino Tarōzaemon who noticed the jellification of a seaweed soup after a night at cold temperatures. It was only in 1859 that the chemist Anselme Payen subjected agar, extracted from the red seaweed *Gelidium corneum*, to chemical analysis [15]. One of the major breakthroughs concerning the use of agar was its utilization as a solid medium for the culture of microorganisms. Although agar was first described for use in microbiology in 1882 by the German microbiologist Walther Hesse, an assistant working in Robert Koch’s laboratory [16], it was Fanny Hesse’s idea, Walther Hesse’s wife, to use agar as an alternative to gelatin. Since then, the knowledge on agar, its extraction processes, and its properties have widely extended.

Agar is a collective term used to describe a mixture of gelling polysaccharides made up of d- and l-galactose. It has a linear sugar skeleton consisting of alternating units of 1-4 linked 3-6-anhydro-α-l-galactose and 1-3-linked β-d-galactopyranose [17]. Agar can be fractionated principally into two components, agarose and agaropectin. Agarose is the gelling fraction, and it is a neutral linear molecule with low levels of sulphation, while agaropectin is the non-gelling fraction characterized as a heterogeneous mixture of smaller molecules including substituted galactose residues and methylated or sulphated sugar units. The relative proportions of agarose and agaropectin are known to vary between species, locality, and environmental conditions. Thus, the selection of raw material can be used to source specific agar functionality [18]. Agar is mainly found in the cell-matrix of red seaweeds of the order Gelidiales (Gelidium and Pterocladia) and Gracilariales (Gracilaria and Hydropuntia), which have become the major worldwide sources [19]. The properties of agar make it suitable for applications in diverse fields with high-added value. For instance, the most common applications include food, feed, cosmetics, pharmaceutical, and biotechnology (mainly microbiology as growth media for culturing bacteria), but potentially emerging applications may also include biomedical, agriculture, biomaterials, and bioplastics.

The industrial production of agar is performed by traditional hot water extraction during several hours under conventional heating [20]. However, this conventional method involves a large volume of water and high-energy requirements; thus, novel strategies using eco-friendly techniques are needed. In this sense, Amparo López-Rubio and her group have made an important contribution, not only for the successful extraction of agar from the red algae *Gelidium sesquipedale* but also for the use of eco-friendly methods following the principles of ‘Green Chemistry’. For instance, Martinez-Sanz et al. reported the production of agar-based extracts by applying ultrasound- and microwave-assisted methods, highlighting the potential of these alternative methods to produce more sustainable agar-based extracts for food-related applications [21,22]. Additionally, a further step in the sustainable use of natural sources by eco-friendly approaches was performed by Sagrario Beltrán’s and Rodrigo Melgosa’s group from the University of Burgos, Spain. They have been working on the valorization of the solid residue generated after agar extraction from the red algae *G. sesquipedale* [23].

In recent years, the use of agar-based biomaterials in emerging areas, such as tissue engineering or ‘smart materials’, has gained great interest. In this sense, the group of Gonçalves and colleagues has made a significant contribution. For instance, Sousa et al. studied the structural and physicochemical properties of agar extracted from *Gracilaria vermiculophylla* under microwave heating [24,25,26]. In a further study, the same research group investigated the formation of agar gels in aqueous media focusing on the different molecular assemblies to understand how these associations can be modified to meet the specific needs of a given application [27]. After a deep characterization of agar and agar gels, they successfully reported the first study of the production of agar-based nanofibers by electrospinning, opening new opportunities for the fabrication of agar-based biomaterials in the form of nanofibers [28].

Another important application for agar-based materials is the production of films for food packaging. Koro De la Caba and her team (University of Basque Country, Spain) have been actively working on the production of food packaging from marine by-products. For instance, they reported the preparation of agar-based films from *G. sesquipedale* using a thermo-molding method to produce renewable and biodegradable films [29,30,31]. Following a similar approach, the group of Gonçalves and colleagues developed different strategies to improve the mechanical strength and water resistance of agar films [32,33] and proposed alternative plasticizers for the production of thermo-compressed agar films [34]. Moreover, the group of Amparo López-Rubio from the Spanish National Research Council presently works on the production of agar-based hydrogels and bioactive aerogels as matrices for the controlled release of bioactive compounds in food systems [35,36]. An original idea from the same group was to use agarose to encapsulate probiotic bacteria and improve their viability during storage, which is a promising approach for the preparation of probiotic functional foods [37].

### 2.2. Chitin and Chitosan

Chitin has a scientific research story of over 200 years. It began in 1811 with the French chemist Henri Braconnot, with chitin isolation and characterization from some fungal species. It was not until 1859 that Rougeut discovered that chitin could be manipulated through chemical and temperature treatments, with heated potassium hydroxide, resulting in a soluble substance. This compound was named “chitosan” in the late 19th century by the German scientist and physiologist Felix Hoppe-Seyler [38]. Since then, researchers efforts have led to the discovery of various sources of chitin, its structural characterization [38,39], and more recently the isolation of nanochitin [40]. In the last two decades, research has focused on improving the properties of chitin- and chitosan-based materials, either through their combination or with other materials or obtaining derivatives by functionalization [41]. Extraction techniques have also been improved, which have evolved from classical chemical extraction of chitin to green chemistry strategies.

Chitin is a semi-crystalline, high-molecular-weight linear polysaccharide made up of β-(1,4)-linked *N*-acetyl-2-amido-2-deoxy-d-glucose units [42]. There are two principal types of crystalline isoforms, α- and β-chitin. Those two isoforms differ on their chain alignment, which determines the final physicochemical properties of the chitin. α-chitin has a compact and highly crystalline structure that results from antiparallel chain arrangement that favors strong hydrogen bonds, whereas the less abundant β-isoform is more reactive and has a high affinity for solvents due to its parallel chain arrangement that provides weaker hydrogen bonds [43]. This biopolymer is present in the cell wall of fungi and is the main compound in the exoskeleton of crustaceans and arthropods and is also present in mollusks, thus making it the second most abundant polysaccharide on earth after cellulose. The major source of chitin is the marine crustacean shells waste from the fisheries industry, such as shrimp, lobster, and crab shells [44].

Rinaudo and colleagues made significant advances on the extraction and characterization of chitin from different marine sources such as α-chitin from shrimp waste and crab shells and β-chitin from cuttlefish bones [43,45,46,47]. They made also a great contribution to the optimization of the deproteination and demineralization processes from shrimp shells. They found that an enzymatic deproteination by crude microbial proteases led to an 88 ± 5% decrease of the protein content, which is similar to the alkali deproteination [48,49,50].

Although chitin presents interesting properties for applications in the biomedical field, such as good biocompatibility, biodegradability, low immunogenicity, and antimicrobial and wound healing activities, its natural insolubility limits its use [51]. Therefore, chitosan, the most important derivative of chitin resulting from its deacetylation [44], has gained great interest [52] because of its higher solubility, the fact that it is positively charged in acidic conditions, presents the unique biological properties of chitin, and has excellent film-forming properties [42,53,54]. Chitosan and its derivatives have great potential in cosmetics, pharmaceutical, and biomedical applications such as delivery systems or bioactive materials for tissue engineering [55,56]. The group of Lina Zhang has developed a large variety of chitin and chitosan-based biomaterials with promising biomedical applications [57,58], particularly hydrogels. Hydrogels that could serve as 3D cell culture platforms [59], cell encapsulation and drug delivery systems [60], and tissue engineering scaffolds [61] were developed by combining the advantageous bioactive properties of chitin and chitosan with other materials. The new techniques developed by this group led to the conception of electroneutral and on-demand dissolvable self-healing hydrogel systems [62,63]. Another important contribution of Zhang et al. was the creation of chitin microspheres based on a chitin solution in a NaOH/urea aqueous system and the use of chitosan microspheres as a sacrificial template [64,65]. This technology has been successfully used in the biomedical field for blood purification therapy [66] and in tissue engineering [67]. Another example of chitosan valorization comes from Eleonora Marsich and colleagues, specialized in carbohydrate polymers. The team focused on the use of lactose-modified chitosan, commercially known as CTL, to mimic biological matrices [68,69,70,71].

In recent years, the use of nanochitin has also gained attention because of its interesting properties at the nano scale, such as high surface area and aspect ratio, mechanical properties, and high antibacterial and anti-inflammatory activities, among others [72]. For instance, the team of Yimin Fan from the Nanjing Forestry University, China, developed new methods to isolate individualized water-dispersed chitin nanofibers and nanocrystals from both α- and β-chitin [73,74]. One of the most promising methodologies was based on the 2,2,6,6-tetramethylpiperidine-1-oxyl radical-mediated oxidation (TEMPO) of chitin followed by mechanical disintegration in water [75]. In a further study, Fan et al. developed a simple pretreatment strategy that improved the oxidation efficiency of TEMPO-mediated isolation and reduced the consumption of the oxidants, resulting in a novel green methodology [76]. Nanochitin can also be employed as nanofiller to obtain reinforced composites of superior mechanical and biological properties [77]. For this purpose, Coltelli and colleagues worked on the use of chitin nanofibrils from fishery biomass to develop bio- and eco-compatible nanocomposites. The incorporation of chitin nanofibrils as a reinforcing agent in extruded composites based on biodegradable polylactic acid (PLA) improved the mechanical properties and provided indirect antimicrobial activity, resulting in potential bioplastics for food packaging and for skin tissue regeneration [78]. Recently, the same effect was performed with cellulose-based bioplastics for food packaging [79]. Another biomedical application of chitin nanofibrils in combination with electronegative nanolignin in microcapsule-like complexes was used to entrap and later deliver both hydrophobic and lipophilic molecules [72].

These are just a few examples of women researchers working on chitin and chitosan. Fortunately, many other women around the world are doing excellent work on this topic and innovating, as is the case of Insiya Jafferjee, who is the co-founder and CEO at Shellworks (https://www.theshellworks.com, accessed on 2 November 2021), a London-based start-up that is developing a method to transform chitin into a novel bioplastic.

### 2.3. Cellulose

#### 2.3.1. Cellulose and Nanocellulose

We have been widely using cellulose as a source of energy and as a material for thousands of years. Nonetheless, its first isolation from plant matter and chemical structure identification was undertaken by the French chemist Anselme Payen in 1838. Since then, multitudinous scientific and technological studies have been made by several scientists around the world on its extraction from different sources, establishment of its chemical and physical structure and morphology, development of different materials from pulp and paper, composites and packaging, to medical materials and high-tech applications, chemoenzymatic modification, cellulose derivatives, new instrumentation, and more recently, isolation of nanocrystals and nanofibers [80,81,82,83]. Herein, we will highlight some works developed by female scientists on (nano)cellulose-based materials from plants and bacteria (Figure 3a–c).

Cellulose is a polysaccharide that is the main constituent of plant cell walls (Figure 3a–c) and the most abundant naturally occurring biopolymer in the biosphere. This linear homo-polysaccharide is composed of repeating β-D-glucopyranose molecules that are covalently linked through acetal functions between the equatorial -OH group of the C4 and of the C1 carbon atom [84]. Cellulose is insoluble in water and in most common solvents due to its strong inter- and intramolecular H-bonding between its individual chain units. Despite its poor solubility, it is used for a wide variety of applications in papermaking, coating, packaging, construction materials, composites, food additives, and in the biomedical fields [80]. Owing to their properties, namely, biocompatibility, biodegradability, nontoxicity and recyclability, and excellent mechanical properties, cellulose and its derivatives such as carboxymethyl cellulose (CMC), cellulose acetate (CA), methyl cellulose (MC), hydroxyethyl cellulose (HEC), and (hydroxypropyl) methyl cellulose (HPMC) have gained a lot of attention [81,82].

In nature, it presents a semicrystalline fibrillar structure of cellulose chains assembled together to form microfibrils, nanofibers, and fibers involved strongly with hemicellulose, lignin, and residual inorganic elements. Using chemical and physical treatments and/or enzymatic-assisted extraction, it is possible to extract cellulose fibers for use in the many application sectors mentioned before [85]. The work developed by Elvira Fortunato and her research group at the New University of Lisbon, Portugal, is an example of one important application of cellulose fibers. They have been working on the development of paper-based transistors as an alternative to silicon-based components. These paper transistors could be used in different applications in daily life such as ‘smart’ packaging, biosensors, animated billboards, and networked shipping labels (Patent: EP2235741, EP2059810) [86,87]. Moreover, these materials are produced at room temperature and are biodegradable, which reduces the negative impact on the environment.

Using top-down approaches, from cellulose fibers, it is possible to isolate nanocellulose, i.e., less than 100 nm in one dimension: cellulose nanofibers (CNF) and cellulose nanocrystals (CNC) [81,82,88]. These nanocellulose forms, aside from the cellulose fiber properties, also exhibit high surface area and aspect ratio, making these cellulose nanoforms very interesting for the development of nanomaterials.

Among the numerous women working on this topic, Kristiina Oksman (Luleå University of Technology, Sweden), Aji Mathew (Stockholm University, Sweden), and Arantxa Eceiza (University of Basque Country, Spain) have achieved significant progresses on nanocellulose. These include: (i) the isolation of nanocellulose (nanofibers, nanocrystals, or whiskers) from different origins and sources such as microcrystalline cellulose from Norway spruce, kenaf fibers, beech pulp, and unbleached rice straw among others by using different isolation approaches, namely, chemical hydrolysis or physical or mechanical isolation methods such as refining, high-pressure homogenization and ultrafine grinder and their characterization [88,89,90,91]; (ii) the processing of functional materials such as nanocomposites with interesting mechanical properties [92,93,94], hydrogels, aerogels [95], and membranes for several applications [96,97]; and (iii) the cellulose-based material development for biomedical applications; for example, as scaffolds [83,98,99].

Particular attention has been focused on cellulose-based nanocomposites presenting interesting mechanical properties prepared using different approaches. For instance, nanocomposites made from cellulose nanofiber with starch powder were investigated by Oksman team. These composites were made using a twin-screw extrusion process. They found that the mechanical properties, as well as the moisture sensitivity of the thermoplastic starch, were improved by preparing the nanocomposites with the cellulose nanofibers [100,101]. Furthermore, polylactic acid-cellulose whisker nanocomposites were synthesized by compounding extrusion. The cellulose whiskers were developed from microcrystalline cellulose and the whiskers were aggregated by strong H-bonds [93]. These whiskers were able to improve the storage modulus of polylactic acid in the plastic region.

Much more work on cellulose-based materials has been and will be carried out by both female and male scientists working together or individually [102,103,104,105].

#### 2.3.2. Bacterial Cellulose

On the other hand, bacterial cellulose (BC, Figure 3a), also known as microbial cellulose, is a naturally occurring 3D network-based material produced as an exopolysaccharide by some aerobic bacteria, such as those from the genus Komagataeibacter. This 3D network is composed of nano- and microfibrils which are 70–80 nm wide and 3–4 nm thick, being 100 times thinner than typical vegetal cellulose fibers [106].

The difference between plant-based cellulose and BC is purity, physicochemical and mechanical properties, and structure. BC is of high purity (free of hemicelluloses and lignin that are usually associated with plant cellulose), crystallinity, and degree of polymerization. Moreover, BC possesses extremely higher water-binding capacity, tensile strength, and surface area, as compared to the widespread plant-based counterparts [107,108,109].

The genus Komagataeibacter is Gram-negative aerobic and non-photosynthetic bacteria capable of converting glucose, glycerol, and other organic substrates into cellulose within a period of a few days in the presence of oxygen. BC can be produced using different bacterial culture media including static, agitated, and bioreactors. Generally, BC production involves expensive culture media. Thus, the use of agroforestry industrial residues could overcome this limitation by serving carbon substrates for the BC production as demonstrated by Carmen S R Freire and collaborators at the University of Aveiro, Portugal [110], or by Arantxa Eceiza and collaborators using pineapple agroindustrial residues [111]. Another study reported on the possibility of using residues from the olive oil production industry as a carbon source for the production of BC by *Gluconacetobacter sacchari* [112]. Furthermore, the by-products of cider, when the apple pomace is mixed with sugar cane, was found to be a potential carbon source for *Gluconacetobacter medellinensis* [113].

When combined with other materials such as alginate, silk fibroin, chitosan, xylans, and starch to form biocomposites, the mechanical and biological properties of bacterial cellulose-based functional materials can be enhanced [7,114,115,116,117]. Cellulose-based functional materials are gaining increasing interest in several industrial fields such as biomedicine, cosmetics, and bioelectronics [109,118].

Carmen S R Freire and collaborators have greatly contributed to the research related to BC-based materials, namely, nanocomposites, membranes, films, etc., by using different approaches such as physical and chemical modification, and polymerization. These materials prepared using mainly BC produced by *Acetobacter xylinum* showed several encouraging properties such as a high mechanical strength and controlled drug loading, making it a promising biopolymer for the production of biomaterials such as optically transparent nanocomposites for different kinds of applications [7,114,115,116,117,119,120,121,122,123]. As mentioned above, they chemically modified the structure of BC to improve its properties. For example, Tome et al. studied BC membranes with tailored surfaces as well as barrier properties for gases using controlled heterogeneous esterification with hexanoyl chloride [124]. Moreover, Fernandes et al. were bio-inspired by the antimicrobial properties of chitosan, and chemically grafted aminoalkyl groups on the BC surface, with the ensuing nanofibrillar network revealing interesting antimicrobial activity and good mechanical properties [116].

Figueiredo et al. from CICECO Laboratory at the University of Aveiro, Portugal, prepared BC-poly(2-hydroxyethyl methacrylate) nanocomposite films by in situ radical polymerization of 2-hydroxyethyl methacrylate, using poly (ethylene glycol) diacrylate as crosslinker. The films thus formed were diaphanous compared to BC and showed improved mechanical performances as well as thermal stability when compared to poly (2-hydroxyethyl methacrylate). As this nanocomposite has proven non-toxicity to human adipose-derived mesenchymal stem cells, it could be used for dry dressing applications [123]. BC-polycaprolactone nanocomposite films were successfully synthesized by incorporating variable amounts of polycaprolactone powder into a BC culture medium. The nanocomposites thus formed could be used for food packaging applications [125].

**Figure 3 polymers-14-01420-f003:**
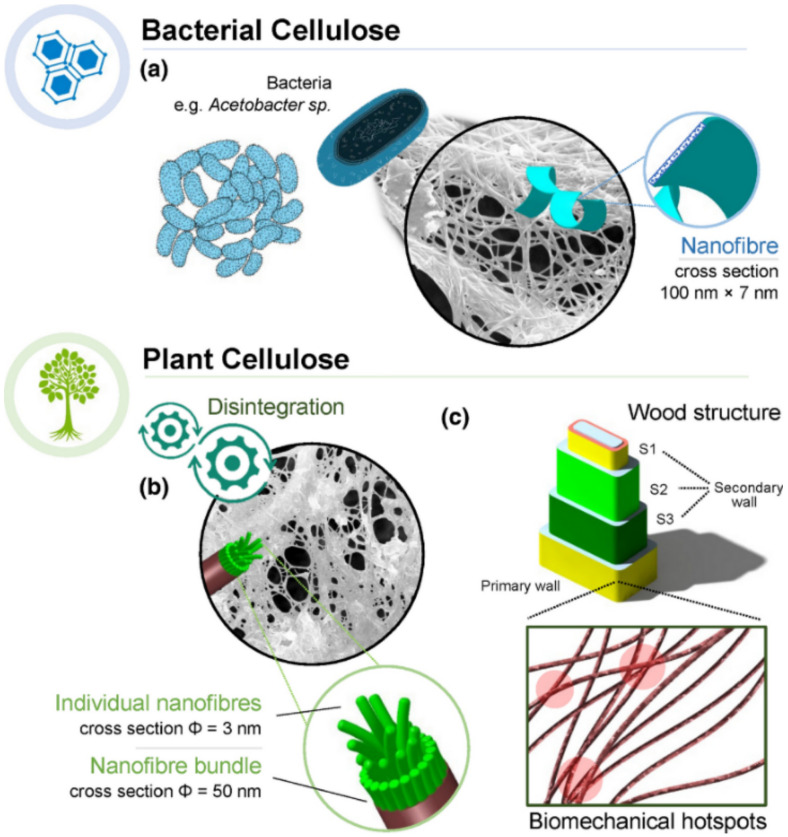
(**a**) Extrusion of a network of bacterial cellulose nanofibrils with associated (nano)fibril cross-sectional morphology. (**b**) Mechanical disintegration of plant matter to produce NFC with associated cross-sectional morphology that corresponds to a bundle of individual cellulose microfibrils. Cellulose microfibrils are present in the cell wall of wood, along with hemicelluloses, protein, and lignin. (**c**) Biomechanical hotspots occur at junctions between two or more microfibrils, or upon close cellulose–cellulose contact, mediated by matrix polysaccharides, such as coiled xyloglucan. Reprinted from ref. [126].

### 2.4. Collagen

From the first use of degraded collagen as a glue more than a hundred years ago to today’s cell therapy, twenty-eight collagen types have been identified and characterized at molecular level [127]. Structural characterization of collagen started in the 1930s and was greatly implemented by researchers such as Ramachandran, the Nobel laureate Crick, Pauling, Rich, and Yonath, and others including Brodsky and Berman [128]. In 1985, Mieczysław Skrodzki, Antoni Michniewicz, and Henryk Kujawa were the first to develop a method to isolate collagen directly from fish skins. In the last 40 years, the research on collagen has greatly increased and has led to today’s collagen therapeutic applications. Collagen is the most abundant fibrous protein found in the connective tissues of a wide range of vertebrates and invertebrate species. It is responsible for providing stability as well as strength to the tissues and thereby gives them their structure [129]. Collagen is a heteropolymer composed of three polypeptide chains in a triple-helical structure. Two of the three chains are identical chains and the remaining one differs in its chemical composition [130]. Collagen has widespread applications in numerous fields, such as pharmaceutical, medical, biomedical, food industry, cosmetics, etc. Majorly, collagen has been used for the cosmetic industry, primarily to increase skin hydration and prevent aging of skin [131].

Collagen has been extracted from various sources, mainly bovine, porcine, and murine animals. Nonetheless, marine organisms are now considered as an interesting alternative source for collagen as they are safer and easier to extract in comparison to terrestrial sources [132]. The traditional extraction method to obtain collagen from terrestrial animal sources is a multi-step process involving the removal of proteins and pigments, a demineralization with HCl or acetic acid, as well as a final digestion using acid or an enzyme [130]. In the eighties, Sylvie Ricard-Blum and her team started working on the biochemical and physicochemical characterization of collagen extracted from fetal calf cartilage. The minor disulfide-bonded collagen was isolated using pepsin treatment [133]. Claire Lethias’s research group from the French National Centre for Scientific Research (CNRS) has been actively working on the isolation of marine collagen from jellyfish such as *Aurelia aurita*, *Cotylorhiza tuberculata*, *Pelagia noctiluca*, and *Rhizostoma pulmo* from the Mediterranean sea coast. The best collagen yield was obtained from *Rhizostoma pulmo*, and this material was found to have applications in the biomedical field, such as for cell adhesion, proliferation, etc. [134,135,136,137,138].

Recently, Sionkowska et al. studied the influence of UV light on the rheological properties of collagen extracted from the skin of the silver carp fish. They found that all the acetic acid collagen solutions they prepared were showing a shear-thinning flow behavior after UV irradiation. They concluded that, depending on the duration of the UV treatment, the collagen could be subjected to photo-degradation or crosslinking. These results were interesting as physically crosslinked collagen can be used for applications in the biomedical, cosmetic, as well as in the food industries [131,139,140].

Table 1 summarizes the origin, extraction methods, and applications of the described biopolymers.

## 3. Discovery of Novel Marine Biopolymers and Carbohydrate-Active Enzymes

### 3.1. Unraveling the Extensive Potential of Polysaccharides from (Marine) Microscopic Life

For the past 20 years, women scientists in natural product research have discovered and characterized several biopolymers from a large diversity of organisms, originating from various ecosystems, and with promising structures and bioactivities. In particular, a large diversity of marine polysaccharides has been described from a plethora of marine organisms (covering all taxonomic domains), enhancing our knowledge of natural polymers from the ocean, and opening up new possibilities of applications in a variety of fields [141]. Moreover, essential work is presently being conducted to increase the visibility of the microscopic life from the ocean since many promising discoveries were attributed to marine bacteria, archaea, and microalgae [142,143]. Whereas numerous and very complete reviewing works were published on the topic of marine polysaccharides [144], we can cite inexhaustively some of the most influential women-led research projects, which enabled important advances in the field of biopolymer discovery.

The microscopic life of the abysses, for instance, has been a great source of inspiration and of novel compound discoveries for the IFREMER team of Sylvia Colliec-Jouault and colleagues [145,146]. Indeed, their research on deep-sea extremophilic microorganisms have led to several discoveries of novel exopolysaccharides (EPS) with interesting structural features and bioactivities. Their significant work on *Alteromonas* strains isolated from deep-sea hydrothermal vents allowed the elucidation and/or chemical modifications of novel promising EPS such as GY785 (Figure 4) [147,148]. These studies pushed them forward to further explore EPS with particular glycosaminoglycan (GAG) features, or GAG-like EPS. Since GAGs are fundamental constituents of both cell surface and extracellular matrix, playing a key role in cell–cell and cell–matrix interactions, they are of particular interest for tissue engineering and repair, as well as for the design and preparation of therapeutic drugs to treat major diseases such as stroke, cancers, and degenerative diseases [149]. In particular, HE800 EPS secreted by *Vibrio diabolicus* (Figure 4), a deep-sea hydrothermal bacterium, was shown to possess a particular hyaluronic acid-like chemical structure and both hexosamines and uronic acids alternating in the repeating unit sequence [150]. It was identified as an effective enhancer for in vivo bone regeneration and to support osteoblastic cell metabolism in culture [151]. Other extreme environments such as hypersaline habitats were explored and led to the description of a novel EPS produced by a halophilic bacterium, *Halomonas stenophila* [152]. The team of Inmaculada Llamas (University of Granada, Spain) behind its discovery named it haloglycan and demonstrated its high viscosity and pseudoplastic behavior, with interesting flocculating, emulsifying, and film-forming activities.

Less extreme but still very promising, cyanoflan, a sulfated carbohydrate, was isolated from a marine cyanobacterium *Cyanothece* sp. and characterized by the team of Paula Tamagnini from i3S, Portugal. They notably demonstrated its high intrinsic viscosity and emulsifying activity in aqueous solutions, making it a promising emulsifying/thickening agent for industrial applications [153]. Its potential application in skin wound healing was also studied and it was shown that it perfectly adapted to the wound bed without inducing systemic or local oxidative or inflammatory reaction [154].

Carol A. M. Nichols from the Commonwealth Scientific and Industrial Research Organisation (CSIRO, Australia) studied EPS-producing bacteria from various ecosystems with a special focus on Antarctic sea ice and the Southern Ocean [155,156]. She also worked on the screening of nearly 800 cultures from the CSIRO Collection of Living Microalgae (CCLM) in search of EPS with potential as adhesives [157].

Recently, a team from the CNRS, based in Roscoff (France), actively participated in the development of strategies for the high-throughput discovery of novel polysaccharides and corresponding carbohydrate-active enzymes (CAZymes) from marine seaweed and bacterial communities, using double-blind techniques such as the Comprehensive Microarray Polymer Profiling (CoMPP) method [158].

### 3.2. Increasing Importance of Marine Carbohydrate-Active Enzymes

Facing the polysaccharide abundance, the seaweeds surrounding microbiomes have adapted themselves to take advantage of this important energy and structural resource. Many bacteria, such as the Zobellia genus, have specialized in marine polysaccharide degradation [9,158] using CAZymes. The CAZymes are currently divided into six large categories: the glycoside hydrolases (GHs) with 172 families, the glycoside transferases (GTs) with 114 families, the polysaccharide lyases (PLs) with 42 families, the carbohydrate esterases (CEs) with 19 families, and 17 families in the auxiliary activities (AA), which are redox enzymes that act in conjunction with CAZymes. The last category is the carbohydrate binding modules (CBMs), non-catalytic proteins that display binding properties to carbohydrates, with 88 inventoried families. CAZymes are listed in the well-maintained CAZy database (http://www.cazy.org, accessed on 5 January 2022). As the targeted marine polysaccharides, marine CAZymes represent huge diversity, with an extremely high potential for biotechnological applications and are mostly under-exploited at the industrial scale until now.

The discovery of CAZymes from a variety of organisms has recently increased with the development of metagenomics and the analysis of large genomic banks. For instance, Marina Isaeva and colleagues from the Russian Academy of Sciences studied the CAZomes (carbohydrate-active enzymes encoded by the genome of an organism) of Zobellia amurskyensis KMM 3526^T^ and Zobellia laminariae KMM 3676^T^ by de novo sequencing and comparative genomics with other strains of the genus. They observed a specialization of this genus for the algal polysaccharides depolymerization in interesting oligosaccharides and monomers with 5.93%, 6.49%, and 6.74% of CAZymes in the predicted coding sequences in Z. laminariae KMM 3676^T^, Z. amurskyensis KMM 3526^T^, and Z. galactanivorans DsiJ^T^, respectively, against 1–5% in most other free living organisms (CAZy database http://www.cazy.org/Genomes.html, accessed on 5 January 2022) [159].

These last few years, genome mining was associated with functional analysis to discover new relevant enzymes, sometimes on complex polysaccharides. Mirjam Czjzek’s team from the CNRS (Roscoff, France) has published very interesting studies of the structural factors governing the catalytic mechanism and substrate specificity of CAZymes, and especially of glycoside hydrolases. For example, they identified several enzymatic activities for the degradation of red algal polysaccharides from Paraglaciecola hydrolytica S66^T^, and novel enzymes degrading the furcellaran, a hybrid carrageenan having both β-carrageenan and κ/β-carrageenan patterns [10]. However, even if the breakthroughs in genome sequencing associated with powerful bioinformatic tools allowed a giant leap in the CAZymes libraries, there is still crucial work to undertake for the functional characterization of the CAZymes. Silvia Vidal-Melgosa and colleagues have been working on this deficit of information, developing a new microarray-based and semiquantitative method to detect whether a polysaccharide was modified or degraded by an enzyme of lysate [160].

Oligosaccharide production is one application of these enzymes that is particularly studied today. As described by Maria Filomena de Jesus Raposo and colleagues in the review about emergent marine prebiotics, oligosaccharides (OS) are part of dietary fibers, which are, according to the Institute of Medicine, a “non-digestible carbohydrate naturally found intact in vegetables/plants” [143,161]. These compounds are not digested by human enzymes and can be metabolized by the intestinal microbiota, with physiological benefits such as modulating the immune system, improving the transit, and regulating the cholesterol level, etc. For example, research led by Yaxuan Sun and colleagues highlighted that chitosan oligosaccharides (COS) have beneficial effects on a model of Alzheimer’s disease [162]. COS are also known for their antimicrobial, anti-inflammatory, and anti-tumor effects. OS are mainly obtained by chemical but also by biochemical (enzymes-assisted) processes. The advantage of the enzymatic method is the production of size-controlled products via an eco-friendly process thanks to the mild conditions of enzymatic reactions and the catalytic specificity of each enzyme. For instance, Ji Young Song and colleagues developed a chitosan digestion process using chitosanases coated on silica gel via glutaraldehyde reticulation to produce COS [163]. The average MW of COS obtained was a function of time, allowing a size control for further applications.

To decipher the degradation mechanism of sulphated polysaccharides, Maria Matard-Mann and colleagues focused one two κ-carrageenases from distant bacteria phylum, both belonging to the GH16 family. By comparing structural and biochemical features, they found some key determinants that would be useful for future applications, such as the synthesis of oligosaccharides [164]. The same research group, led by Mirjam Czjzek, worked on many other marine polysaccharides; for example, an agar-specific hydrolase from the algal-associated bacterium Z. galactanivorans, able to degrade natural agar, which presents complex patterns (as complex as carrageenans) [165]. The team also succeeded in purifying and characterizing different oligosaccharides released by the action of the recombinant ZgAgaC on Osmundea pinnatifida agar. Colliec Jouault’s team also worked on the depolymerization of the EPS secreted by Alteromonas infernus (GY785), by setting a screening of 26 commercial enzymes as well as a lysate of the cell producing the GY785. While the commercial enzymes could not depolymerize the EPS, the lysate could, showing that the bacteria contained the enzymes capable of degrading its EPS [8,9], in biologically interesting oligosaccharides. This work showed the great potential of microorganisms to develop biotechnological applications for carbohydrate modifications. Numerous other teams around the world work on marine CAZymes to develop polysaccharide degradation processes.

To sum up, the growing interest for marine polysaccharide processing via enzymatic ways is driven by many reasons, including the increasing interest for marine oligosaccharide benefits, and the need for green and eco-friendly processes. The development of enzymatic processes for marine polysaccharide degradation, the understanding of the catalytic mechanism of the associated CAZymes, and the discovery of original CAZymes have a bright future, of which women are an integral part.

## 4. Conclusions and Perspectives

The dedication of women scientists around the world during the last 20 years has led to several discoveries and innovations within the field of biopolymer science and technology. These important advances settle the premises of a promising future in biopolymer-related research and development.

As an alternative to synthetic polymers, biopolymers are playing a key role in the frame for the circular bioeconomy. Thus, there are other important subjects in the field of biopolymers that must be taken into account in future studies such as the sustainable use of natural resources, influence on biodiversity, new green extraction approaches, and the end use of the new bio-based materials, i.e., biodegradation, recyclability, or reuse.

To finalize, the present contributions will be paving the way for the next generation of female and male scientists on this topic.

## Figures and Tables

**Figure 1 polymers-14-01420-f001:**
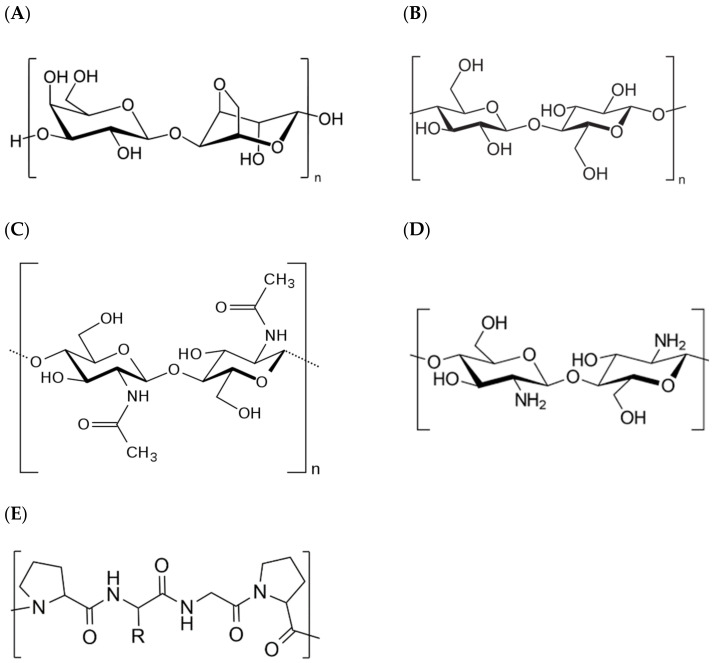
Chemical structure of (**A**) agarose, (**B**) cellulose, (**C**) chitin, (**D**) chitosan, and (**E**) collagen.

**Figure 2 polymers-14-01420-f002:**
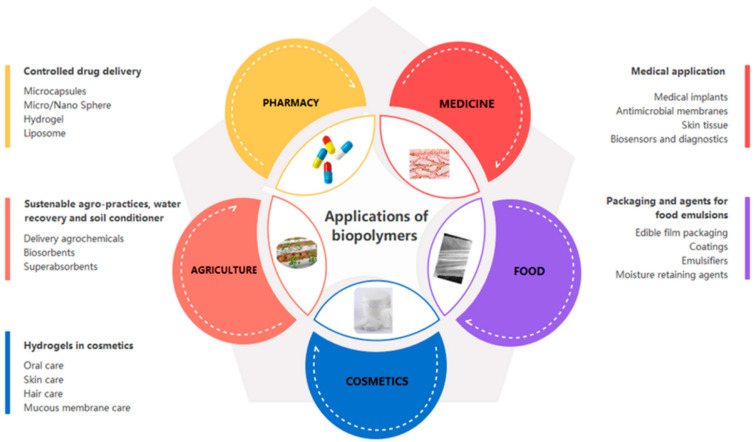
Applications of biopolymers. Reprinted from ref. [14].

**Figure 4 polymers-14-01420-f004:**
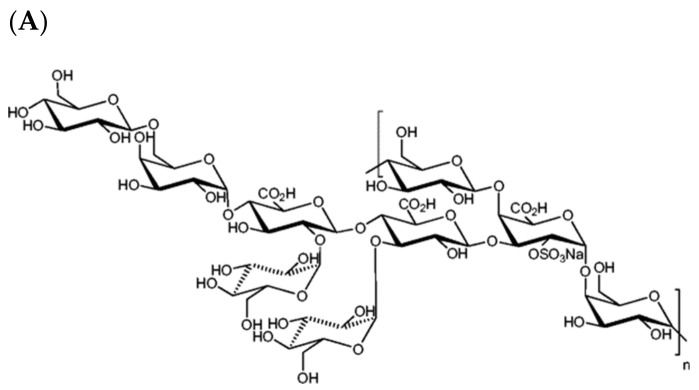
Chemical structure of marine exopolysaccharides: (**A**) GY785, and (**B**) HE800 EPS (adapted from refs. [147,148,150]).

**Table 1 polymers-14-01420-t001:** Biopolymers’ origin, extraction methods, and applications.

Biopolymers	Origin and Extraction Methods	References
*Agar*	From red seaweeds of the order Gelidiales and Gracilariales	
	*Extraction methods:*	
	- hot water extraction for several hours	[20]
	- ultrasound assisted methods	[21,22]
	- microwave-assisted methods	[24,25,26]
*(Nano)Chitin*	From the exoskeleton of crustaceans and arthropods, and from mollusks	
	*Chitin extraction methods:*	
	- conventional methods with acid and alkaline solutions	[43,45,46,47]
	- enzymatic extraction	[48,49,50]
	*Nanochitin extraction methods:*	
	- nanocrystals by TEMPO-mediated isolation + mechanical disintegration in water	[74,75,76]
*(Nano)Cellulose*	From plants and wood	
	*Cellulose extraction methods:*	
	- chemical and physical treatments and/or enzymatic-assisted extraction of cellulose	[85]
	*Nanocellulose extraction methods:*	
	- chemical hydrolysis, high-pressure homogenization, ultrafine grinder isolation methods for nanoforms	[88,89,90,91,94]
*Bacterial Cellulose*	From aerobic bacteria—genus Komagataeibacter	
	- production by static, agitated and bioreactors culture media	[110,111,112,113]
*Collagen*	From animal cartilage, fish skin, jellyfish	
	*Extraction methods:*	
	- traditional multi-step process with acetic acid or HCl and enzymes	[130,132,133]
**Biopolymers**	**Applications**	**References**
*Agar*	- agar-based nanofibers by electrospinning	[28]
	- agar-based films for food packaging	[29,30,31,32,33]
	- agar-based hydrogels and bioactive aerogels	[35,36]
*(Nano)Chitin*	- carboxymethyl chitin hydrogel	[59]
	- chitin hydrogels with self-healing property	[60]
	- chitin/carbon nanotubes composite hydrogels	[61]
	- chitosan-chitin nanofiber composites	[77]
	- chitin nanofibrils in materials for packaging	[78]
*(Nano)Cellulose*	- paper-based transistors	[86,87]
	- (nano)composites	[90,91,92,93,94,100]
	- cellulose nanofiber aerogel	[95]
	- 3D printed porous cellulose hydrogel scaffolds	[95]
*Bacterial Cellulose*	- BC-based biocomposites	[114,115,116,117]
	- transparent nanocomposites	[119,120,121,122,123]
	- nanocomposite films by in situ radical polymerization	[123,125]
*Collagen*	- collagen-based materials for cosmetic applications	[131]
	- collagen gels	[136]

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
