# Peer review of "Contributions of Women in Recent Research on Biopolymer Science"

_polymers, 2022, doi:10.3390/polym14071420_

Round 1

Reviewer 1 Report

Dear Editor,

Thank you for providing me an opportunity study the manuscript titled “Women in Polymer Science and Technology: Focus on the Advances on Biopolymers” The article found is interesting. Thus, the scope of the manuscript is adequate to the topics of this Journal. It is recommended to receive this article with the following changes.

1.Prof. Alina Sionokowska (https://orcid.org/0000-0002-1551-2725), worked extensively in the field of Polymeric biomaterials, such as (i) Chitosan , (ii) Collagen, (iii), (iv)plastic-mimetic polypeptide. The reviewers may consider her work and cite the articles.

  1. Mrs Kathyayani(https://orcid.org/0000-0001-6744-742X) published few articles on miscibility characteristics, thermal stability of peptide-based polymers, authors can discuss on the peptide-derived polymers in brief and cite their articles also.
  2. Reviewers can add a table in consolidating the work on collagen, polypeptdes chitosan etc done by woman scientists.

Author Response

Reviewer 1 :

Dear Editor,

Thank you for providing me an opportunity study the manuscript titled “Women in Polymer Science and Technology: Focus on the Advances on Biopolymers” The article found is interesting. Thus, the scope of the manuscript is adequate to the topics of this Journal. It is recommended to receive this article with the following changes.

  1. Prof. Alina Sionokowska (https://orcid.org/0000-0002-1551-2725), worked extensively in the field of Polymeric biomaterials, such as (i) Chitosan , (ii) Collagen, (iii), (iv)plastic-mimetic polypeptide. The reviewers may consider her work and cite the articles.

Dear Reviewer, thank you for your suggestion. We agree that Prof. Alina Sionokowska has published great works on biopolymers. Several of her papers are cited in our paper in the collagen part.

  1. Mrs Kathyayani(https://orcid.org/0000-0001-6744-742X) published few articles on miscibility characteristics, thermal stability of peptide-based polymers, authors can discuss on the peptide-derived polymers in brief and cite their articles also.

Thank you for your suggestion, however, we decided not to add a part on peptide-derived polymers on this short review. Mrs Kathyayani’s work is however interesting and could indeed be include in a review on this subject.

  1. Reviewers can add a table in consolidating the work on collagen, polypeptdes chitosan etc done by woman scientists.

Dear reviewer, we agree that a table would have been a good addition on a review about biopolymers. However, our objective for this paper is to highlight female scientist on the topic.

Reviewer 2 Report

The manuscript entitled Women in Polymer Science and Technology: Focus on the Ad- 2 vances on Biopolymers by Unnimaya Thalakkale Veettil et al highlights the work done by women in the field of Polymer science.

The aim of this paper is to underline women-led research in the field of polymer science during last 20 years.

I think that this manuscript should be rejected because of the following reasons:

Major reasons –

  1. A review of work on biopolymers by women in last 20 years is not useful from the scientific point of view as a scientist from polymer science would not read this review and get an idea about biopolymers on general. He/she would only learn about some topics, cherry-picked to fit the ‘women in biopolymer science’ issue, and miss out on a lot of information in general.
  2. This review could be shaped in a way that highlights breakthroughs by women in biopolymer science (mind you, there are plenty). Talking about the work of women in the past 20 years overlooks the very purpose of this issue, in my opinion.
  3. Moreover, this review leaves an impression that women, in fact, require special attention and care. This is not true. We need to promote women in science, while maintaining a standard for them in the scientific society. If we only pick work by one gender in a review paper, it fails science as a whole. In my opinion, this paper is framed in an inappropriate manner. The essence of the paper should be to shed light on the unknown, like ‘Critical Perspective: Named Reactions Discovered and Developed by Women’ by JULIE A. OLSON AND KEVIN M. SHEA*. Such an article really makes you think about women in science.
  4. The biopolymers explained in this paper are not appropriately explained. A more detailed explanation is required.

Minor Reasons -

  1. While introducing Agar, it is a must to mention Fanny Hesse, who first suggested to use Agar as an alternative to gelatin.
  2. Line 174 – ‘These are just a few examples of women researchers working on chitin and chitosan.’ This is not true. There are many more women working on chitin and chitosan like Agnieszka Sobczak-merchant, Bozena Tyliszczak, nihal ozel, to name a few.

To conclude, I would like to suggest a re-submission of this paper after incorporating the suggestions.

Author Response

Reviewer 2 :

The manuscript entitled ‘Women in Polymer Science and Technology: Focus on the Ad- 2 vances on Biopolymers’ by Unnimaya Thalakkale Veettil et al highlights the work done by women in the field of Polymer science.

The aim of this paper is to underline women-led research in the field of polymer science during last 20 years.

I think that this manuscript should be rejected because of the following reasons:

Major reasons –

  1. A review of work on biopolymers by women in last 20 years is not useful from the scientific point of view as a scientist from polymer science would not read this review and get an idea about biopolymers on general. He/she would only learn about some topics, cherry-picked to fit the ‘women in biopolymer science’ issue, and miss out on a lot of information in general.
  2. This review could be shaped in a way that highlights breakthroughs by women in biopolymer science (mind you, there are plenty). Talking about the work of women in the past 20 years overlooks the very purpose of this issue, in my opinion.
  3. Moreover, this review leaves an impression that women, in fact, require special attention and care. This is not true. We need to promote women in science, while maintaining a standard for them in the scientific society. If we only pick work by one gender in a review paper, it fails science as a whole. In my opinion, this paper is framed in an inappropriate manner. The essence of the paper should be to shed light on the unknown, like ‘Critical Perspective: Named Reactions Discovered and Developed by Women’ by JULIE A. OLSON AND KEVIN M. SHEA*. Such an article really makes you think about women in science.
  4. The biopolymers explained in this paper are not appropriately explained. A more detailed explanation is required.

Dear reviewer, thank you for your insight on this paper, especially because we apparently did not build it in a way that was beneficial for the researchers as well as the reader. Our aim with this manuscript was to do an introductive paper to this special issue by highlighting some works on biopolymers done by female researchers. Our initial idea was to do a commentary-like manuscript and not a review.

Thank you for your interesting suggestion to read Julie Olson and Kevin Shea paper. It was indeed very interesting. Taking in account your suggestions, we modified the introduction of the paper, some clumsy words in the abstract and we added necessary introduction of the historical discovery on the biopolymers and we hope that our attempt to write a commentary for the women in science issue does not give the wrong message. Moreover, I also modified the title of the manuscript.

Minor Reasons -

  1. While introducing Agar, it is a must to mention Fanny Hesse, who first suggested to use Agar as an alternative to gelatin.

Thank you very much for your suggestion. We agree with your suggestion and mentioned Fanny Hesse great idea on our paper.

  1. Line 174 – ‘These are just a few examples of women researchers working on chitin and chitosan.’ This is not true. There are many more women working on chitin and chitosan like Agnieszka Sobczak-merchant, Bozena Tyliszczak, nihal ozel, to name a few.

Dear reviewer, maybe our sentence was clumsy as we agree that we could not cite all the names of women working on chitin and tried to express that in this sentence.

To conclude, I would like to suggest a re-submission of this paper after incorporating the suggestion

Reviewer 3 Report

I am an female with a long-standing record of working in teams of mixed genders. I do see the problems with participarion of women in academic communities, but feel that it is equalizing now.

This is well written and organized paper, however, I feel uneasy to review it.

The idea of the whole volume was to deal with the problem  "...that important gender differences still persist in academia today." I do understand that it is very ambitious task to write good and critical papers on that subject.

The paper of Prof. Fernandes (initiator of the volume) et al., however, is constructed in such a way that describes the four major groups of biopolymers (and addiionaly marine polysaccharides as separate paragraph) with indication - "look out some women-scientist are also active on this field". In my opinion it is not good layout of the paper, mostly  from the point of view of women. Of course I have no idea how improve the paper and if the Editors will decide that paper is o.k. and in line with the idea of the volume I do accept its publication but I would propose to rethink that.

Author Response

Reviewer 1 :

Dear Editor,

Thank you for providing me an opportunity study the manuscript titled “Women in Polymer Science and Technology: Focus on the Advances on Biopolymers” The article found is interesting. Thus, the scope of the manuscript is adequate to the topics of this Journal. It is recommended to receive this article with the following changes.

  1. Prof. Alina Sionokowska (https://orcid.org/0000-0002-1551-2725), worked extensively in the field of Polymeric biomaterials, such as (i) Chitosan , (ii) Collagen, (iii), (iv)plastic-mimetic polypeptide. The reviewers may consider her work and cite the articles.

Dear Reviewer, thank you for your suggestion. We agree that Prof. Alina Sionokowska has published great works on biopolymers. Several of her papers are cited in our paper in the collagen part.

  1. Mrs Kathyayani(https://orcid.org/0000-0001-6744-742X) published few articles on miscibility characteristics, thermal stability of peptide-based polymers, authors can discuss on the peptide-derived polymers in brief and cite their articles also.

Thank you for your suggestion, however, we decided not to add a part on peptide-derived polymers on this short review. Mrs Kathyayani’s work is however interesting and could indeed be include in a review on this subject.

  1. Reviewers can add a table in consolidating the work on collagen, polypeptdes chitosan etc done by woman scientists.

Dear reviewer, we agree that a table would have been a good addition on a review about biopolymers. However, our objective for this paper is to highlight female scientist on the topic.

Reviewer 2 :

The manuscript entitled ‘Women in Polymer Science and Technology: Focus on the Ad- 2 vances on Biopolymers’ by Unnimaya Thalakkale Veettil et al highlights the work done by women in the field of Polymer science.

The aim of this paper is to underline women-led research in the field of polymer science during last 20 years.

I think that this manuscript should be rejected because of the following reasons:

Major reasons –

  1. A review of work on biopolymers by women in last 20 years is not useful from the scientific point of view as a scientist from polymer science would not read this review and get an idea about biopolymers on general. He/she would only learn about some topics, cherry-picked to fit the ‘women in biopolymer science’ issue, and miss out on a lot of information in general.
  2. This review could be shaped in a way that highlights breakthroughs by women in biopolymer science (mind you, there are plenty). Talking about the work of women in the past 20 years overlooks the very purpose of this issue, in my opinion.
  3. Moreover, this review leaves an impression that women, in fact, require special attention and care. This is not true. We need to promote women in science, while maintaining a standard for them in the scientific society. If we only pick work by one gender in a review paper, it fails science as a whole. In my opinion, this paper is framed in an inappropriate manner. The essence of the paper should be to shed light on the unknown, like ‘Critical Perspective: Named Reactions Discovered and Developed by Women’ by JULIE A. OLSON AND KEVIN M. SHEA*. Such an article really makes you think about women in science.
  4. The biopolymers explained in this paper are not appropriately explained. A more detailed explanation is required.

Dear reviewer, thank you for your insight on this paper, especially because we apparently did not build it in a way that was beneficial for the researchers as well as the reader. Our aim with this manuscript was to do an introductive paper to this special issue by highlighting some works on biopolymers done by female researchers. Our initial idea was to do a commentary-like manuscript and not a review.

Thank you for your interesting suggestion to read Julie Olson and Kevin Shea paper. It was indeed very interesting. Taking in account your suggestions, we modified the introduction of the paper, some clumsy words in the abstract and we added necessary introduction of the historical discovery on the biopolymers and we hope that our attempt to write a commentary for the women in science issue does not give the wrong message. Moreover, I also modified the title of the manuscript.

Minor Reasons -

  1. While introducing Agar, it is a must to mention Fanny Hesse, who first suggested to use Agar as an alternative to gelatin.

Thank you very much for your suggestion. We agree with your suggestion and mentioned Fanny Hesse great idea on our paper.

  1. Line 174 – ‘These are just a few examples of women researchers working on chitin and chitosan.’ This is not true. There are many more women working on chitin and chitosan like Agnieszka Sobczak-merchant, Bozena Tyliszczak, nihal ozel, to name a few.

Dear reviewer, maybe our sentence was clumsy as we agree that we could not cite all the names of women working on chitin and tried to express that in this sentence.

To conclude, I would like to suggest a re-submission of this paper after incorporating the suggestion

Reviewer 3 :

I am an female with a long-standing record of working in teams of mixed genders. I do see the problems with participarion of women in academic communities, but feel that it is equalizing now.

This is well written and organized paper, however, I feel uneasy to review it.

The idea of the whole volume was to deal with the problem  "...that important gender differences still persist in academia today." I do understand that it is very ambitious task to write good and critical papers on that subject.

The paper of Prof. Fernandes (initiator of the volume) et al., however, is constructed in such a way that describes the four major groups of biopolymers (and addiionaly marine polysaccharides as separate paragraph) with indication - "look out some women-scientist are also active on this field". In my opinion it is not good layout of the paper, mostly  from the point of view of women. Of course I have no idea how improve the paper and if the Editors will decide that paper is o.k. and in line with the idea of the volume I do accept its publication but I would propose to rethink that.

Dear reviewer, thank you for your insight on this paper, especially because we apparently did not build it in a way that was beneficial for the researchers as well as the reader. As we described in the commentaries of the second reviewer, our aim with this manuscript was to do an introductive paper to this special issue by highlighting some works on biopolymers done by female researchers. As already mentioned, our initial idea was to do a commentary-like manuscript and not a review.

Taking in account your suggestions, we have modified the paper, some clumsy words in the abstract and we added necessary introduction of the historical discovery on the biopolymers and we hope that our attempt to write a commentary for the women in science issue does not give the wrong message. Moreover, I also modified the title of the manuscript.

Reviewer 4 :

The following few specific comments are suggested to improve the manuscript.

  1. What is the motive to choose this title, “Women in Polymer Science and Technology: Focus on the advances on Biopolymers” The authors should give the literature review curve on the published work which men and women have published on the biopolymers.

Dear reviewer, thank you for your suggestion. We added an introduction on the historical discoveries for each biopolymer, from female and male studies.

  1. The introduction of this review paper should be rewritten clearly, and the role of women in this research area should be addressed in the introduction.

Dear reviewer, we agree with you on the necessity to modify our introduction. We changed it and hope it is now clearer.

  1. What is the novelty of this review paper compared to a published review article on biopolymers?

Dear reviewer, the aim of this paper is not to have a ‘novel and original’ review on biopolymers. Our aim with this manuscript was to do an introductive paper to this special issue by highlighting some works on biopolymers done by female researchers.

  1. For easy understanding, the authors must add overview figures and/or tables to compare the process/extraction yield % of biopolymers in this review.

Dear reviewer, many thanks for your comment. Nonetheless, our intention with this pqper is not to do a the aim of this paper is not to have a ‘novel and original’ review on biopolymers.

  1. The future perspective should be addressed in this review paper.
  2. The conclusion of this manuscript should be extended.

Dear reviewer, thanks for your suggestions. We have added a sentence on the conclusion on a general perspective on the topic of this review.

Reviewer 4 Report

The following few specific comments are suggested to improve the manuscript.

  1. What is the motive to choose this title, “Women in Polymer Science and Technology: Focus on the advances on Biopolymers” The authors should give the literature review curve on the published work which men and women have published on the biopolymers.
  2. The introduction of this review paper should be rewritten clearly, and the role of women in this research area should be addressed in the introduction.
  3. What is the novelty of this review paper compared to a published review article on biopolymers?
  4. For easy understanding, the authors must add overview figures and/or tables to compare the process/extraction yield % of biopolymers in this review.
  5. The future perspective should be addressed in this review paper.
  6. The conclusion of this manuscript should be extended.

Author Response

Reviewer 4 :

The following few specific comments are suggested to improve the manuscript.

  1. What is the motive to choose this title, “Women in Polymer Science and Technology: Focus on the advances on Biopolymers” The authors should give the literature review curve on the published work which men and women have published on the biopolymers.

Dear reviewer, thank you for your suggestion. We added an introduction on the historical discoveries for each biopolymer, from female and male studies.

  1. The introduction of this review paper should be rewritten clearly, and the role of women in this research area should be addressed in the introduction.

Dear reviewer, we agree with you on the necessity to modify our introduction. We changed it and hope it is now clearer.

  1. What is the novelty of this review paper compared to a published review article on biopolymers?

Dear reviewer, the aim of this paper is not to have a ‘novel and original’ review on biopolymers. Our aim with this manuscript was to do an introductive paper to this special issue by highlighting some works on biopolymers done by female researchers.

  1. For easy understanding, the authors must add overview figures and/or tables to compare the process/extraction yield % of biopolymers in this review.

Dear reviewer, many thanks for your comment. Nonetheless, our intention with this pqper is not to do a the aim of this paper is not to have a ‘novel and original’ review on biopolymers.

  1. The future perspective should be addressed in this review paper.
  2. The conclusion of this manuscript should be extended.

Dear reviewer, thanks for your suggestions. We have added a sentence on the conclusion on a general perspective on the topic of this review.

Round 2

Reviewer 2 Report

The manuscript titled 'Recent advancements in biopolymer science: examples of research projects led by women' is well suited for the issue, however, this paper would not be very useful for a scientist pursuing research in biopolymers because the data is selective. But since the issue requires this kind of selective data, it can be published in the present form.

Author Response

The manuscript titled 'Recent advancements in biopolymer science: examples of research projects led by women' is well suited for the issue, however, this paper would not be very useful for a scientist pursuing research in biopolymers because the data is selective. But since the issue requires this kind of selective data, it can be published in the present form.

Dear reviewer, thank you for your comments.

Reviewer 3 Report

As previously stated I do not feel happy about that manuscript - anyway if it is an ntroduction to the volume could be o.k.

Author Response

As previously stated I do not feel happy about that manuscript - anyway if it is an introduction to the volume could be o.k.

Dear reviewer, thank you for your comments.

Reviewer 4 Report

The authors did not reply appropriately to most of the comments in the revised manuscript—for example; 1. The authors should give the literature review curve on the published work that men and women have published on the biopolymers if this review paper aims to expose the contribution of women in biopolymer science. Then, the literature review curve should be added to the revised manuscript.  

  1. For easy understanding, the authors must add overview figures and/or tables to compare the process/extraction yield % of biopolymers in this review. The author stated in reply to comment 4, “Nonetheless, our intention with this paper is not to do the aim of this paper is not to have a ‘novel and original’ review on biopolymers.” If this review paper does not have novelty and originality, so what were the authors' aims to write this review paper? 3. In reply to these two comments (5. The future perspective should be addressed in this review paper. And 6. The conclusion of this manuscript should be extended.), just the authors have completed the formality. The future perspective should be added separately in this review paper.
  2. The title of this review paper should be modified as “Contribution of women in recent research in biopolymer science.”
  3. Kindly add the references in lines 55-67 and 223-232.

Author Response

The authors did not reply appropriately to most of the comments in the revised manuscript—for example;

  1. The authors should give the literature review curve on the published work that men and women have published on the biopolymers if this review paper aims to expose the contribution of women in biopolymer science. Then, the literature review curve should be added to the revised manuscript.  

Dear reviewer, thank you for your suggestion, it is a good idea. Nonetheless, we did not find the good research filters to do such a curve. Moreover, taking in account the number of publications on biopolymers and the difficulty to identify the female and male first names in some languages, we regret but it is very complicate to respond to your request.

  1. For easy understanding, the authors must add overview figures and/or tables to compare the process/extraction yield % of biopolymers in this review.

Dear reviewer, thank you for your suggestion. We have added a Table summarizing the origins, extraction processes and applications of the biopolymers presented and some general illustrations in the revised manuscript.

  1. The author stated in reply to comment 4, “Nonetheless, our intention with this paper is not to do the aim of this paper is not to have a ‘novel and original’ review on biopolymers.” If this review paper does not have novelty and originality, so what were the authors' aims to write this review paper?

Dear reviewer, our aim with this manuscript was to do an introductive paper to this special issue (we have described it in the abstract) by highlighting some works on biopolymers done by female researchers. Our initial idea was to do a commentary-like manuscript and not a review.

  1. In reply to these two comments (5. The future perspective should be addressed in this review paper. And 6. The conclusion of this manuscript should be extended.), just the authors have completed the formality. The future perspective should be added separately in this review paper.

Dear reviewer, we have added the perspectives together with the conclusion, but in a separated paragraph in the revised manuscript.

  1. The title of this review paper should be modified as “Contribution of women in recent research in biopolymer science.”

Dear reviewer, we have changed the title as suggested by you in the revised manuscript.

  1. Kindly add the references in lines 55-67 and 223-232.

Dear reviewer, we have added the references in the revised manuscript.

Round 3

Reviewer 4 Report

The authors need to revise the carefully recent version review paper. The reviewer observed a lot of scientific and other errors in the revised manuscript. The following few specific comments are suggested to improve the manuscript.  

  1. The authors stated, “The aim of this Special Issue is to highlight the work designed and developed by women on biopolymer science and technology. Thus, this short review aims to do an introduction to this Special Issue by highlighting some recent contributions of women around the world on the particular topic of biopolymer science and technology during the last 20 years.” Is it a special issue or review paper? It should be rephrased.
  2. In lines 351-357, Are these lines figure caption of figure 3? What about the figure caption of figure 3c? The address of Figures 3a, 3b & 3c should be mentioned in the text of the revised manuscript.
  3. In lines 400 and 403, the two table captions are addressed’ for example, “Table 1 summarizes the origin, extraction methods and applications of the described 400 biopolymers. Table 1. Biopolymers origin, extraction methods and applications.” Which one is correct?
  4. Figure 2 on page 15, Is it figure 2 or figure 4?  

Author Response

Reviewer 4 :

The authors need to revise the carefully recent version review paper. The reviewer observed a lot of scientific and other errors in the revised manuscript. The following few specific comments are suggested to improve the manuscript.  

  1. The authors stated, “The aim of this Special Issue is to highlight the work designed and developed by women on biopolymer science and technology. Thus, this short review aims to do an introduction to this Special Issue by highlighting some recent contributions of women around the world on the particular topic of biopolymer science and technology during the last 20 years.” Is it a special issue or review paper? It should be rephrased.

Dear reviewer, many thanks for your comment. A Special Issue is a collection of articles that concentrates on a topical research area within the scope of a journal. In the present Special Issue, the topic is ‘Women in Polymer Science and Technology: Biopolymers’. The present manuscript is a short review on this topic, that has as objective to introduce the Special Issue, since the corresponding author is one of the Guest Editors.

The sentence was rephrased.

2. In lines 351-357, Are these lines figure caption of figure 3? What about the figure caption of figure 3c? The address of Figures 3a, 3b & 3c should be mentioned in the text of the revised manuscript.

Dear reviewer, many thanks for your comment. Figure 3c is now mentioned in the revised manuscript.

3. In lines 400 and 403, the two table captions are addressed’ for example, “Table 1 summarizes the origin, extraction methods and applications of the described 400 biopolymers. Table 1. Biopolymers origin, extraction methods and applications.” Which one is correct?

Dear reviewer, both are correct:

Table 1 summarizes the origin, extraction methods and applications of the described biopolymers. Is the sentence that introduce Table1!

Table 1. Biopolymers origin, extraction methods and applications. Is the title of Table1!

4. Figure 2 on page 15, Is it figure 2 or figure 4?  

Dear reviewer, many thanks for your comment. The number of the figure was corrected in the revised manuscript.